# ViroGym: Realistic Large-Scale Benchmarks for Evaluating Viral Proteins

## Abstract

Protein language models (pLMs) have shown strong potential in prediction of the functional effects of missense variants in zero-shot settings. Despite this progress, benchmarking pLMs for viral proteins remains limited and systematic strategies for integrating in silico metrics with in vitro validation to guide antigen and target selection are underdeveloped. Here, we introduce ViroGym, a comprehensive benchmark designed to evaluate variant effect prediction in viral proteins and to facilitate selecting rational antigen candidates. We curated 79 deep mutational scanning (DMS) assays encompassing eukaryotic viruses, collectively comprising 552,937 mutated amino acid sequences across 7 distinct phenotypic readouts, and 21 influenza virus neutralisation tasks and a real-world predictive task for SARS-CoV-2. We benchmark well-established pLMs on fitness landscapes, antigenic diversity, and pandemic forecasting to provide a framework for vaccine selection, and show that pLMs selected using in vitro experimental data excel at predicting real-world viral evolution.

## 1 Introduction

The most clinically relevant respiratory viruses–such as influenza, SARS-CoV-2, and others–mutate at a rapid pace, challenging both the immune system and development of effective vaccines and treatments. Even with extensive near real-time genomic reporting systems, such as GISAID Shu & McCauley (2017) and Nextstrain Hadfield et al. (2018), people are often having to anticipate as to the future direction of these rapidly evolving pathogens, with mismatches between the predicted and actual trajectory resulting both public health and individual consequences.

A familiar example is the current vaccine development system for SARS-CoV-2 and influenza, which involves a semi-annual strain selection process recommended by the World Health Organization (WHO). This production system, especially for seasonal influenza vaccines, has remained largely unchanged for over 40 years Wei et al. (2020). Moreover, the effectiveness of seasonal influenza vaccines from 2009 to 2025 flu seasons is only in the range of 19%-60% Centers for Disease Control and Prevention and others (2025), and the peak vaccine effectiveness for SARS-CoV-2 in autumn 2023 is 50.6% within 2-4 weeks but then dropped sharply to 13.6%, largely due to the emergence of new variants Kirsebom et al. (2024). Despite of suboptimal vaccine efficacy, manufacturers must produce and release vaccines within six months of WHO announcements.

Given the need to design, pilot, manufacture, and test vaccines against emerging strains, a proactive vaccine design framework is needed to enable scientists to initiate preparation for manufacturing prior to WHO strain announcements. The ideal framework should also be broad enough to cover viruses associated with infectious diseases, such as Zika virus, Hepatitis B virus, and Human Immunodeficiency Virus (HIV). With the proven success of large language models (LLMs), it is plausible that such a proactive framework could be effective.

LLMs trained to predict amino acid sequences, known as pLMs, have had success with estimating the functional impact and fitness consequences of candidate mutations without requiring prior evolutionary or epidemiological information Meier et al. (2021), demonstrating its great potential in enabling early-stage anticipation of antigenic changes and supporting proactive vaccine design. While current pLMs have largely been validated on non-viral sequences, with most of the foundational model training explicitly masking viral sequences from training, testing, and validation sets.

Therefore, there remains a gap in our understanding of how different pLMs perform with viral genomic sequences. A clear set of benchmarks relevant to modelling of viral evolution is a key step towards applying pLM to vaccine and antiviral development.

To address these limitations, we present ViroGym, a realistic large-scale benchmark designed to evaluate pLMs in zero-shot settings for global vaccine development. The benchmark consists of three core tasks:

- **Mutational effect prediction**, which evaluates model ability to capture complex, non-linear correlations within viral genomic sequences and to infer the functional consequences of individual mutations.

- **Antigenic diversity prediction**, which assesses model capacity to understand immune escape and strain differentiation.

- **Pandemic prediction**, which identifies models with strong zero-shot generalization suitable for modelling mutations observed in natural viral evolution.

ViroGym includes over 552,937 mutated sequence readouts, 2,691 viral sequence–titer pairs, and 24,187 naturally occurring single-mutation frequency measurements. It spans 13 virus types and 7 phenotypic categories, providing broad coverage across viral families and functional properties (see Table 7 in Appendix A.1 for details). By providing clinical meaningful and rigorous benchmarks, ViroGym enables a more realistic assessment of model utility for vaccine and antiviral development.

## 2 RELATED WORK

**ProteinGym.** ProteinGym is a benchmark suite designed to evaluate pLMs on their ability to predict the functional effects of protein mutations. It aggregates large-scale DMS datasets across a wide range of proteins, mutation types, and functional assays and defines biologically grounded evaluation metrics Notin et al. (2023). The majority of the prediction tasks involve non-viral proteins, with 24 out of 217 assays derived from viral sequences.

**EVEREST.** EVEREST evaluates pLMs performance on viral mutational fitness prediction using a curated benchmark of 45 viral DMS datasets and finds that current pLMs fail to reliably predict mutations for over half of these viruses Gurev et al. (2025). Because its primary focus is on priority viruses, many other available viral DMS assays are not included in the benchmark.

**DMS Correlation Studies.** Livesey and Marsh Livesey & Marsh (2025) recently collected 13 new DMS datasets from ProteinGym and evaluated 97 variant effect predictors (VEPs) across 36 human proteins. They observed a strong correspondence between VEP performance on DMS benchmarks and their ability to classify clinical variants, particularly for predictors not trained on clinical data. These findings suggest that VEPs could complement, and in some cases partially substitute for, in vitro experiments in assessing variant effects.

## 3 VIROGYM

The benchmark comprises 79 DMS assays, 21 sequencing-based neutralisation assays for influenza A, and a real-world prediction task derived from the Global Initiative on Sharing All Influenza Data (GISAID), which provides genomic surveillance data for SARS-CoV-2.

Figure 1 illustrates the overall framework. Pre-trained pLMs on large sequence databases such as UniProtKB and BFD are evaluated by computing the in silico score for each amino acid sequence using suitable scoring strategies. The evaluation is divided into two main components: the DMS and neutralization assays serve as in vitro experimental prediction tasks, while the GISAID dataset enables real-world pandemic prediction. We assess model performance by comparing in silico scores against both experimental measurements and naturally occurring mutations, providing a large-scale benchmark across controlled and real-world settings.

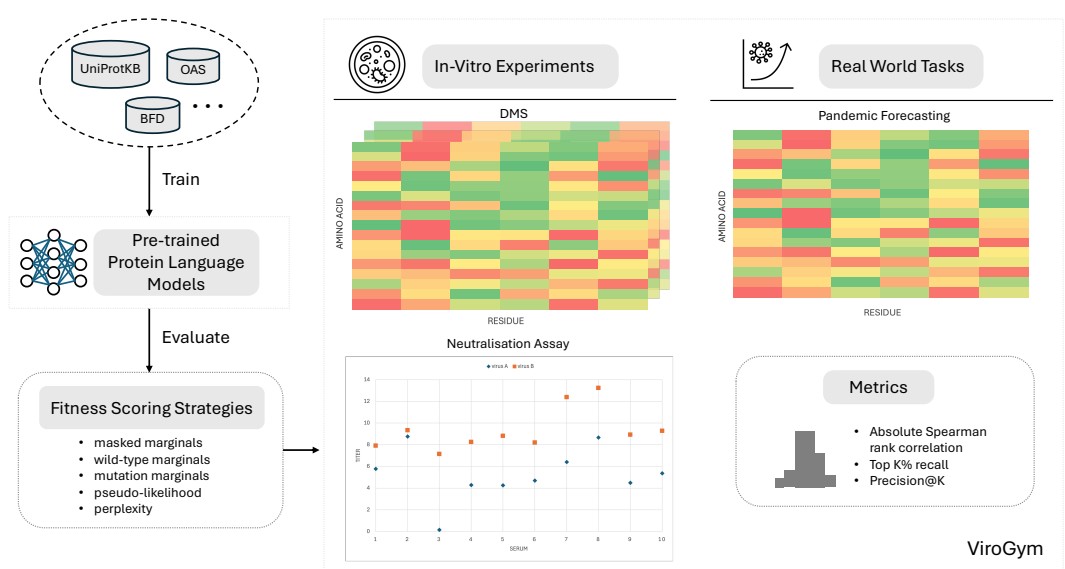

Figure 1: ViroGym benchmark framework. The benchmark consists of two major components: in vitro experimental evaluation and real-world prediction tasks. The in vitro evaluation leverages experimental measurements from DMS assays and neutralisation assays to evaluate model performance on protein functional effects. The real-world component evaluates models on SARS-CoV-2 pandemic forecasting using viral sequence data from GISAID database, capturing model generalisation from controlled wet lab settings to natural viral evolution.

## 3.1 DATASET SOURCES

**DMS.** DMS is a high-throughput experimental technique that characterizes a protein's functional landscape by systematically evaluating millions of its single–amino-acid variants and mapping each mutation (genotype) to a measured functional property (phenotype) Fowler & Fields (2014). The selection of DMS assays in ViroGym follows the guidelines established by ProteinGym. As a result, ViroGym includes DMS assays covering SARS-CoV-2 Starr et al. (2020; 2022b;a); Taylor & Starr (2023; 2024); Dadonaite et al. (2024b; 2025a), Influenza A Welsh et al. (2024); Dadonaite et al. (2024a); Yu et al. (2025), HIV Haddox et al. (2016; 2018); Radford et al. (2023); Radford & Bloom (2025) and 10 other viruses (Detailed reference can be found in Table 5 and 6 of Appendix A.1). Beyond the functional categories considered in ProteinGym, ViroGym introduces an additional function type: immune escape, which represents a critical phenotype for viral proteins and is directly relevant to vaccine and therapeutic development.

**Neutralisation assay.** In contrast to traditional serological assays, which assess antibody neutralisation against a single viral strain per serum sample, sequence-based high-throughput neutralisation assays quantify serum antibody using neutralisation titers across all relevant viral strains within a single experiment Loes et al. (2024) (see Table 8 of Appendix A.1 for details). This dense, sequence-resolved measurement paradigm enables machine learning models jointly learning over viral sequence variation and antigenic response. As a result, such models can understand predictive mappings between viral evolution and antibody-mediated immunity, facilitating the identification of antigenicity novel epitopes.

**GISAID database.** GISAID is a global surveillance platform that monitors priority pathogens and facilitates the sharing of their genetic sequences and associated metadata Shu & McCauley (2017). This resource enables researchers to track viral evolution and transmission dynamics during epidemics and pandemics.

## 3.2 DATASET PROCESS

**DMS.** To faithfully reflect the underlying protein function experiments, we collected the corresponding target sequence for each DMS assay, following the guidelines established by ProteinGym. For certain SARS-CoV-2 functional assays that evaluate only the receptor-binding domain (RBD), we truncated the Spike protein sequence to the assayed region. Immune escape phenotypic assays typically do not disclose detailed information about the vaccine formulation or serum source. Consequently, for these assays, we aggregate measurements by averaging DMS scores across different sera evaluated against the same viral sequence.

**Neutralisation assay.** We reviewed published information for patients participating in the neutralisation assay experiments to manually curate their vaccination histories. From this, we identified the specific vaccines each patient had received and obtained the corresponding HA1 sequences. This allowed us to accurately assess the antigenic coverage provided by these vaccines.

**GISAID database.** We collected all circulating SARS-CoV-2 sequences from the GISAID database spanning January 1, 2020, to May 31, 2025. From these sequences, we extracted all mutations in the Spike protein and recorded their observed occurrences. The resulting heat map, which depicts the actual prevalence of each mutation including deletions at each residue, is shown in heatmap$_G ISAID$.

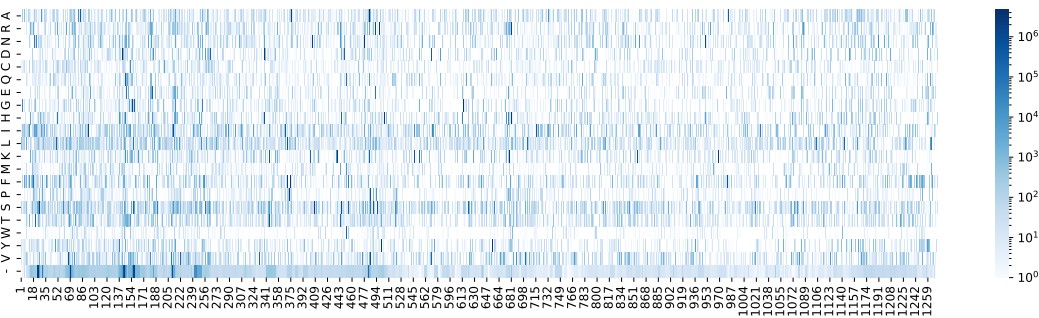

Figure 2: SARS-CoV-2 Spike Protein Mutation Heat Map. This heat map displays the frequency of 21 potential amino acid substitutions across 1273 residues of the SARS-CoV-2 Spike protein, with colour intensity indicating mutation frequency at each position. Data were collected from the GISAID database between January 2020 and May 2025.

## 3.3 BASELINES

Similar to how language models learn grammar and contextual meaning from text, pLMs can learn biological rules and functional properties from amino acid sequences. Leveraging the rapid growth of protein sequence data, researchers have trained pLMs using unsupervised learning to generate representations that capture information ranging from protein structure to biochemical properties, providing features for a wide array of biomedical applications Rives et al. (2021).

In this work, we focus on single-sequence pLMs representative of current approaches – ESM-1 Rives et al. (2019) as the first generation of pLMs; ESM-1v Meier et al. (2021) enabling zero-shot variant fitness prediction; ESM-2 Lin et al. (2023) is available in model sizes ranging from 8M to 15B parameters; ProtT5 Elnaggar et al. (2022) is an encoder-decoder architecture designed to capture contextual meaning in amino acid sequences, whose embeddings support models such as VESPA and VESPAI Marquet et al. (2022); ProGen2 suite Nijkamp et al. (2023) exploring dataset and scale effects, spanning antibody-specific models to the large BFD90-trained mode; ProtGPT2 Ferruz et al. (2022) aimed at de novo protein generation; Tranception Notin et al. (2022) achieving robust performance at modelling the fitness landscape of protein sequences.

Our primary focus is on the ability of pLMs to predict variant fitness accurately in a zero-shot setting, as determining precise protein function experimentally can take weeks or months. For example,

during the COVID-19 pandemic, structural analysis revealing atomic-level conformations of the SARS-CoV-2 RBD was completed one month after the first full genome sequences were available Wrapp et al. (2020). Therefore, given the time efficiency of early vaccine development and data leakage risks, we mainly focus on single sequence-based pLMs in ViroGym.

## 3.4 Evaluation metrics

We adopt ranking-based metrics throughout ViroGym to identify high-impact mutations for practical use.

**Absolute Spearman rank correlation.** Spearman's rank correlation coefficient is used across all prediction tasks, as it is well suited for evaluating agreement between predicted and experimentally measured rankings Notin et al. (2023). This metric is particularly relevant for applications such as vaccine strain selection, where correctly ranking mutational effects is more critical than predicting their absolute values.

**Top K% recall.** Experimental measurements inevitably contain noise, which disproportionately affects the ranking of low-impact mutations. To mitigate this effect, we focus on the top K% of mutations ranked by experimental measurements and report recall within this subset. Following established convention from ProteinGym, we use the top 10% recall metric consistently across all prediction tasks in ViroGym.

**Precision@K.** Precision@K is introduced in the pandemic forecasting task to provide a complementary view of model performance, measuring the accuracy of identifying high-risk variants among the top K predictions.

## 4 Results

### 4.1 Mutational effect prediction

The concept of leveraging language models to predict protein function in a zero-shot setting was first introduced by Meier et al. Meier et al. (2021), who systematically compared four scoring methods for evaluating mutational effects: masked marginals, wild-type marginals, mutation marginals, and pseudo-likelihood. Their analyses showed that the masked marginals approach outperformed the others and has subsequently been adopted in ESM-2 for predicting mutational effects on protein fitness.

However, when we evaluated encoder-based models on the DMS experiments from ViroGym using these four strategies, we observed no significant performance differences. One alternative approach is to leverage the contextual embeddings generated by the language model to compute a similarity metric, analogous to sentence similarity in natural language processing. In this framework, we quantify how far a mutated sequence drifts from its reference sequence. We found that Euclidean distance (defined in Equation 1) works better in general.

$$d(wildtype, variant) = \|\bar{\mathbf{h}}_{wildtype}^{(L)} - \bar{\mathbf{h}}_{variant}^{(L)}\|_2 \tag{1}$$

where $\bar{\mathbf{h}}^{(L)}$ is the mean pool of the contextual embedding from the last hidden layer $L$. Euclidean distance provides more accurate predictions of mutational impact than cosine similarity, as is more sensitive to single-point mutations in long sequences.

Other pLMs with decoder-only architectures employ different scoring strategies. For example, ProGen2, Tranception, and ProtGPT2 rely primarily on negative log-likelihood or perplexity score, while DeepSequence Riesselman et al. (2018) and MULAN Frolova et al. (2025) use a likelihood ratio-based approach to mitigate biases from local sequence context. Beyond these strategies, some researchers draw an analogy to natural language, considering the probability of observing a mutant at a specific position as a measure of evolutionary grammaticality Hie et al. (2021); Allman et al. (2025), reflecting how plausible a mutation is in the protein sequence context.

The question of which in silico scoring method most effectively represents protein function remains open, particularly as pLMs grow in sophistication and application. Understanding this question will be pivotal for translating large language models from predictive tools into mechanistic frameworks capable of guiding experimental protein design.

Table 1: Performance of ESM2-650M under different scoring strategies. Results are reported as the average top 10% recall and absolute Spearman's rank correlation between model predictions and experimental measurements.

| STRATEGY | RECALL | STD. | SPEARMAN | STD. |
|---|---|---|---|---|
| MASKED | 0.1144 | 0.046 | 0.1091 | 0.1209 |
| WILDTYPE | 0.1125 | 0.0443 | 0.1065 | 0.1159 |
| MUTATION | 0.1147 | 0.0434 | 0.1087 | 0.1183 |
| GRAMMAR | 0.1250 | 0.0458 | 0.1151 | 0.1145 |
| SEMANTIC | **0.1375** | 0.0612 | **0.1693** | 0.1034 |
| RATIO | 0.0813 | 0.0444 | 0.1092 | 0.1127 |
| LOSS | 0.1044 | 0.0564 | 0.1205 | 0.1261 |

Table 2: Zero-shot performance on the DMS benchmark. Results are reported as the average top 10% recall and absolute Spearman's rank correlation between model scores and experimental measurements across all baselines.

| MODEL | RECALL | STD. | SPEARMAN | STD. |
|---|---|---|---|---|
| VESPAL | 0.1635 | 0.0726 | 0.2715 | 0.1402 |
| VESPA | 0.1702 | 0.0796 | 0.2797 | 0.1506 |
| TRANCEPT. | 0.1572 | 0.0681 | 0.2271 | 0.1300 |
| PROTGPT2 | 0.1105 | 0.0370 | 0.1021 | 0.0732 |
| PROGEN2 | **0.1980** | 0.0910 | **0.2930** | 0.1583 |
| ESM1V | 0.1451 | 0.0644 | 0.1877 | 0.1026 |
| ESM1 | 0.1466 | 0.0586 | 0.1997 | 0.1021 |
| ESM2 | 0.1419 | 0.0639 | 0.1741 | 0.1122 |

Based on our experimental results in Table 1, we conclude that when the wild-type amino acid sequence is available, the most effective strategy is to compare the semantic changes between the mutated and reference sequences. This approach aligns well with both machine learning principles and biological interpretation, providing a robust method for identifying high-impact mutations.

The overall results for mutational effect prediction task are presented in Table 2, with detailed performance metrics are reported in Appendix A.2 (Table 9, Figure 6 and 7). ProGen2 achieves the strongest performance, as illustrated in Figure 3, which shows per-task results using ESM2 15B as an example. However, the remaining models show no statistically significant performance differences.

## 4.2 ANTIGENIC DIVERSITY PREDICTION

The current influenza vaccine strains are selected based on the degree to which circulating viruses have drifted from previously dominant strains, with this distance assessed by integrating both genetic and antigenic evolution Smith et al. (2004); Fouchier & Smith (2010). While modelling genetic evolution from historical data is routine - typically via phylogenetic trees constructed using maximum likelihood estimation (MLE) Felsenstein (1973), capturing antigenic evolution still requires wet-lab experimental inputs. If a pLM can predict whether an emerging strain is likely to be covered by a given vaccine strain, it could significantly accelerate the vaccine development cycle.

To evaluate this capability, we established 21 influenza neutralisation assays to measure the ability of pLMs to detect antigenic differences among viral strains. These assays generally use haemagglutination inhibition techniques, in which antibody titers serve as a proxy for antigenic similarity - for example, a titer of 1:40 is often considered indicative of adequate immune coverage Hannoun et al. (2004). Within this framework, we query pLMs to estimate the antigenic similarity between vaccine strains and newly isolated viral strains. Conceptually, higher similarity scores should correspond to stronger expected vaccine-mediated protection.

To quantitatively assess model performance, we calculate the contextual embedding distance between a circulating strain and the vaccine strain as the predicted antigenic distance and evaluate its correlation with experimental titers. For decoder-only models, we use perplexity as the predicted

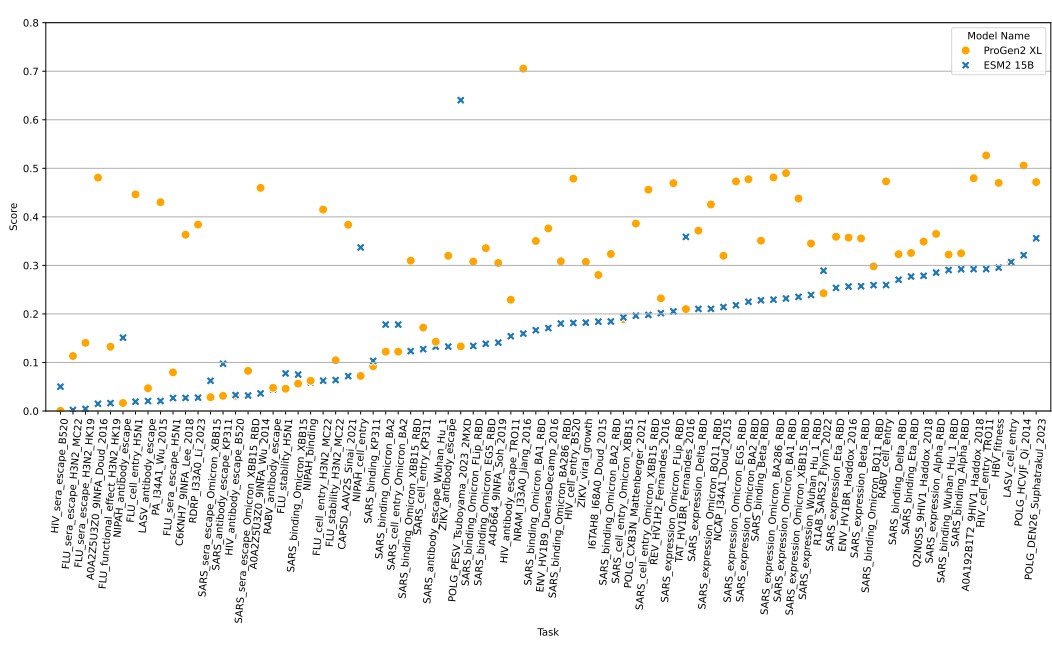

Figure 3: Task-wise comparison of ESM2 15B and ProGen2-XL on the DMS benchmark. ESM2 15B scores are computed using the semantic scoring strategy, while ProGen2-XL scores use the negative log-likelihood strategy. Reported values represent the absolute Spearman's rank correlation between model fitness scores and experimental measurements.

Table 3: Zero-shot neutralisation prediction results. Reported values are the average absolute Spearman's rank correlation between model predictions and experimental measurements across all baseline methods.

| Model Name | Spearman | Std. |
|---|---|---|
| ProtT5 | 0.1961 | 0.206 |
| Tranception | **0.2316** | 0.1696 |
| ProtGPT2 | 0.2018 | 0.1845 |
| ProGen2 | 0.2250 | 0.1852 |
| ESM1v | 0.2282 | 0.2043 |
| ESM1 | 0.2222 | 0.2098 |
| ESM2 | 0.2267 | 0.1840 |

antigenic distance. This evaluation enables us to determine whether a pLM can approximate fine-grained antigenic relationships and provide actionable immunological insights. Thus, by accurately ranking strains in terms of antigenic similarity, pLMs could guide vaccine strain selection to maximize coverage against circulating viruses and optimize antibody-mediated protection.

However, the performance differences among the models are marginal, with Tranception M slightly outperforming the others in Table 3 (see task-wise performance in Appendix A.2 Table 8). Detailed performance for Tranception M on each task can be found in Figure 4. These results suggesting that current pLMs exhibit similar capabilities on neutralisation prediction tasks and that significant room for improvement remains.

## 4.3 Pandemic prediction

pLMs are increasingly viewed as a universal key for protein prediction, potentially replacing traditional multiple sequence alignment (MSA) methods Weissenow & Rost (2025). Their capabilities

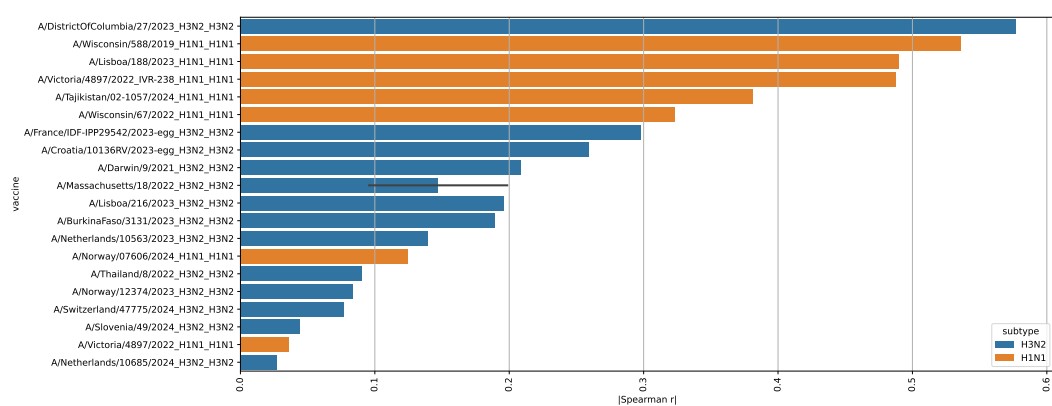

Figure 4: Task-wise performance of Tranception M on the neutralisation benchmark. Antigenicity scores are computed using the negative log-likelihood strategy. Each task corresponds to a vaccine strain representing post-vaccination serum, with colors indicating influenza A subtypes. Performance is measured as the absolute Spearman's rank correlation between predicted antigenicity scores and experimental measurements, averaged across sera from different animal sources.

include generating representations for secondary and tertiary structure prediction and inferring biochemical properties without labelled data Rives et al. (2021), identifying conserved residues without MSAs Marquet et al. (2022), and predicting the effects of missense mutations Meier et al. (2021). Many pLMs have demonstrated outstanding performance on in vitro benchmarks, but a critical question remains: can these models generalize effectively to in vivo or real-world environments?

To address this, we designed an evaluation task to project in vitro results onto real-world scenarios. Specifically, we test whether pLMs can identify dominant circulating mutations using only the target SARS-CoV-2 Spike protein sequence. Each model calculates the fitness score of every single mutation using either semantic scoring strategies (e.g., ESM-1 family, ESM-1v, ESM-2 family) or perplexity-based scoring (e.g., VESPA, VESPAI, Tranception family, ProTGPT2, ProGen2 suite). Heat maps for all baselines of predicting the in silico fitness score for each amino acid per residue can be found in Appendix A.2 Figures 9-16.

To quantify performance, we introduce a precision@K metric, which measures model ability to correctly identify the top mutations. Across three evaluation metrics, ProGen2-XL shows the strongest achievement in this task showing in Table 4. Notably, it is also dominating mutation-level prediction tasks (Table 2) and achieving reasonable performance on the neutralisation task (Table 3).

Next, we investigate the relationships among computational predictions, in vitro experiments, and real-world viral evolution, with the aim of assessing how effectively pLMs can bridge laboratory assays and naturally circulating viral strains. To this end, we focus on SARS-CoV-2 and analyse the overlap of single-point mutations identified under comparable experimental conditions.

We observe that ProGen2-XL, which achieves the best overall performance across our benchmarks, shares nearly 50% of the top-ranked mutations with those most prevalent in real-world viral circulation in Figure 5. In contrast, DMS assays identify only 10% of these dominant circulating mutations. It is worth noting that ProGen2-XL exhibits approximately 20% overlap with the top mutations identified by DMS and among these shared mutations is the N501Y substitution, which has been shown to be a major determinant of the increased transmissibility of the SARS-CoV-2 Alpha variant by enhancing the binding affinity of the Spike protein to host cell receptors Liu et al. (2022).

These findings suggest that, although DMS assays characterize protein fitness under controlled conditions, appropriately selected pLMs can more effectively capture evolutionary constraints that govern viral spread in real-world settings.

Table 4: Zero-shot pandemic prediction results. Metrics reported for all baselines include Top 10% Recall, absolute Spearman's rank correlation, and Precision@3 between model scores and mutation frequencies from GISAID.

| MODEL NAME | RECALL | SPEARMAN | PRECISION |
|---|---|---|---|
| VESPAL | 0.3226 | 0.3602 | 0 |
| VESPA | 0.2936 | 0.3152 | 0 |
| TRANCEPTION | 0.2066 | 0.2196 | 0 |
| PROTGPT2 | 0.1221 | 0.0372 | 0 |
| PROGEN2 | **0.4081** | **0.3153** | **0.33** |
| ESM1V | 0.1206 | 0.0543 | 0 |
| ESM1 | 0.2471 | 0.1870 | 0 |
| ESM2 | 0.2608 | 0.3026 | 0 |

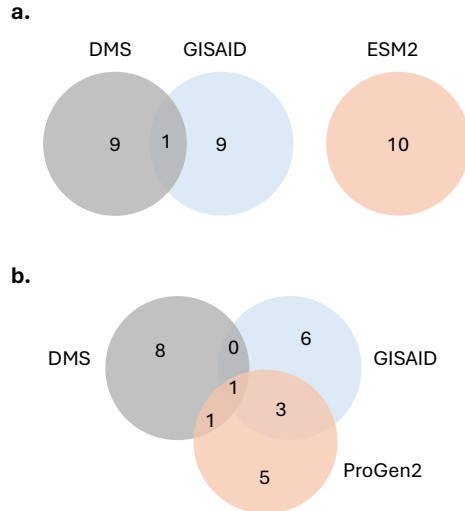

Figure 5: Overlap among top 10 mutations from computational predictions, in vitro DMS assays of the SARS-CoV-2 S protein RBD, and naturally occurring mutations from GISAID. **a.** ESM2-650M predictions show no overlap with DMS or GISAID mutations. **b.** Predicted mutations from ProGen2-XL overlap 50% with GISAID and 20% with DMS.

## 5    DISCUSSION

ViroGym introduces a novel evaluation framework for viral proteins, encompassing mutational effect prediction, antigenicity diversity prediction, and pandemic prediction, with the goal of linking in vitro experiments to real-world outcomes. While DMS datasets provide a detailed view of the protein fitness landscape by measuring protein properties under controlled conditions, protein evolution in real-world is shaped by additional constraints, particularly for viral proteins. For instance, immune imprinting from early antigen exposure can bias antibody responses toward conserved epitopes, influencing vaccine strain selection and subsequently shaping viral evolutionary trajectories.

Our analysis highlights three key considerations for improving pLMs on viral proteins. A key challenge for pLMs is to handle insertions and deletions (indels), which often disrupt protein function. Currently, only ESM models, to our knowledge, explicitly encode deletions as tokens in their vocabulary, and filtering out sequences with deletions yields modest performance gains across models (see in Appendix A.2 Table 10). Secondly, unlike other sequence-based pLMs trained solely on UniProtKB, the best performing model in ViroGym - ProGen2 might benefit from joint pretraining on UniProtKB and BFD datasets. Lastly, viral proteins frequently exceed the typical length of proteins, whereas most pLMs are limited to context windows of fewer than 1024 residues. These

observations suggest that pLMs could achieve improved performance by expanding their token representations, incorporating larger and more diverse training datasets, and increasing context length to better capture long-range dependencies in viral proteins.

ViroGym is the first benchmark to systematically integrate computational models, experimental assays, and real-world viral evolution data, providing a unified platform for evaluating model predictive performance and guiding vaccine development. Our results demonstrate that, although DMS assays identify the top mutations based on experimental fitness, these mutations do not substantially overlap with real-world circulating variants. Interestingly, models selected using DMS-based evaluation successfully predict the dominant mutations observed in natural viral circulation. This indirect validation indicates that the mutation effect prediction task in ViroGym serves as a useful proxy for identifying models that capture biological constraints generalizable to real-world viral evolution. At the same time, it suggests that DMS assays may have limited utility for fine-tuning pLMs, given their low overlap with circulating variants.

Importantly, our results also indicate that DMS and pLM-based predictions provide complementary signals. While DMS assays offer high resolution measurements of functional effects under well-defined experimental conditions, pLMs capture broader sequence-level constraints learned from large-scale evolutionary data. Combining DMS-derived fitness information with pLM predictions may therefore enable more accurate and robust forecasting of real-world viral evolution, offering a promising direction for improving mutation prioritization and vaccine strain selection.

## 6   LIMITATIONS AND FUTURE WORK

MSA-based models like EVE Frazer et al. (2021) and EVEscape Thadani et al. (2023) are excluded because high-quality multiple sequence alignments are often difficult to obtain for viral proteins, particularly for novel viruses. We also do not consider hybrid models and structural-based models such as MULAN because they rely on experimentally validated protein structures, which are time-consuming to obtain. While tools such as AlphaFold Jumper et al. (2021) can predict complex protein structures rapidly, a systematic comparison of predictions based on AlphaFold versus experimentally solved structures remains an important direction for future work in vaccine design.

A limitation of our work is that pandemic prediction task focuses primarily on top mutations from SARS-CoV-2, as reliable mutation frequency data for Influenza A and other viruses are more difficult to obtain from GISAID. Extending this task to additional viral species would enable a more thorough evaluation of model generalization.

## IMPACT STATEMENT

This work highlights an important shift in how pLMs can be evaluated and applied: rather than merely reproducing outcomes from DMS experiments, pLMs may be better suited to capture real-world mutagenic patterns observed during natural viral evolution. By benchmarking models against experimentally grounded and naturally occurring mutations, our framework suggests that pLMs can provide more relevant and actionable insights for real-world applications such as vaccine design, surveillance, and therapeutic development. This perspective supports the use of pLMs as complementary tools to experimental assays, with the potential to guide and prioritize future experimental efforts.

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

# A APPENDIX

## A.1 BENCHMARK CONSTRUCTION DETAILS

Table 5: Sources of DMS benchmark datasets. Mutational effect prediction tasks are based on DMS assays. This table summarises the data resources from publicly available datasets.

| Dataset | Phenotype | Reference |
|---|---|---|
| ZIKV | viral growth | Sourisseau et al. (2019) |
| ZIKV | immune escape | Sourisseau et al. (2019), Kikawa et al. (2023) |
| RABV | immune escape | Aditham et al. (2025) |
| RABV | cell entry | Aditham et al. (2025) |
| NIPAH | binding | Larsen et al. (2025) |
| NIPAH | immune escape | Larsen et al. (2025) |
| NIPAH | cell entry | Larsen et al. (2025) |
| LASV | immune escape | Carr et al. (2024) |
| LASV | cell entry | Carr et al. (2024) |
| HIV B520 | immune escape | Radford et al. (2023) |
| HIV B520 | cell entry | Radford & Bloom (2025) |
| HIV B520 | immune escape | Radford et al. (2023), Radford & Bloom (2025) |
| HIV TRO11 | cell entry | Radford & Bloom (2025) |
| HIV TRO11 | immune escape | Radford & Bloom (2025) |
| *HIV HXB2 | viral growth | Fernandes et al. (2016) |
| *HIV BRU/LAI | viral growth | Fernandes et al. (2016) |
| *HIV strain896 | viral growth | Duenas-Decamp et al. (2016) |
| *HIV BRU/LAI | viral growth | Haddox et al. (2016) |
| *HIV | viral growth | Haddox et al. (2018) |
| *HIV B520 | viral growth | Haddox et al. (2018) |
| HBV | fitness | Yu et al. (2024) |
| *SCV2 RBD Wuhan hu | binding | Starr et al. (2020) |
| *SCV2 RBD Wuhan hu | expression | Starr et al. (2020) |
| SCV2 RBD Alpha | binding | Starr et al. (2022a) |
| SCV2 RBD Alpha | expression | Starr et al. (2022a) |
| SCV2 RBD Beta | binding | Starr et al. (2022a) |
| SCV2 RBD Beta | expression | Starr et al. (2022a) |
| SCV2 RBD Delta | binding | Starr et al. (2022a) |
| SCV2 RBD Delta | expression | Starr et al. (2022a) |
| SCV2 RBD Eta | binding | Starr et al. (2022a) |
| SCV2 RBD Eta | expression | Starr et al. (2022a) |
| SCV2 RBD Omicron BA.1 | binding | Starr et al. (2022b) |
| SCV2 RBD Omicron BA.1 | expression | Starr et al. (2022b) |
| SCV2 RBD Omicron BA.2 | binding | Starr et al. (2022b) |
| SCV2 RBD Omicron BA.2 | expression | Starr et al. (2022b) |
| SCV2 RBD Omicron BQ.1.1 | binding | Taylor & Starr (2023) |
| SCV2 RBD Omicron BQ.1.1 | expression | Taylor & Starr (2023) |
| SCV2 RBD Omicron XBB.1.5 | binding | Taylor & Starr (2023) |
| SCV2 RBD Omicron XBB.1.5 | expression | Taylor & Starr (2023) |
| SCV2 RBD Omicron XBB.1.5 | binding | Taylor & Starr (2023) |
| SCV2 RBD Omicron XBB.1.5 | expression | Taylor & Starr (2023) |
| SCV2 RBD Omicron BA.2.86 | binding | Taylor & Starr (2024) |
| SCV2 RBD Omicron BA.2.86 | expression | Taylor & Starr (2024) |

*represents the dataset is from ProteinGym

Table 6: Sources of DMS benchmark datasets (continued). Mutational effect prediction tasks are based on DMS assays. This table summarises the data resources from publicly available datasets.

| Dataset | Phenotype | Reference |
|---|---|---|
| SCV2 RBD Omicron EG.5 | binding | Taylor & Starr (2024) |
| SCV2 RBD Omicron EG.5 | expression | Taylor & Starr (2024) |
| SCV2 RBD Omicron FLip | binding | Taylor & Starr (2024) |
| SCV2 RBD Omicron FLip | expression | Taylor & Starr (2024) |
| SCV2 Wuhan hu | immune escape | Cao et al. (2022) |
| SCV2 RBD Omicron XBB.1.5 | immune escape | Dadonaite et al. (2025a) |
| SCV2 RBD Omicron XBB.1.5 | cell entry | Dadonaite et al. (2025a) |
| SCV2 Omicron XBB.1.5 | immune escape | Dadonaite et al. (2024b) |
| SCV2 Omicron XBB.1.5 | binding | Dadonaite et al. (2024b) |
| SCV2 Omicron XBB.1.5 | cell entry | Dadonaite et al. (2024b) |
| SCV2 Omicron BA.2 | binding | Dadonaite et al. (2024b) |
| SCV2 Omicron BA.2 | cell entry | Dadonaite et al. (2024b) |
| SCV2 KP.3.11 | immune escape | Dadonaite et al. (2025b) |
| SCV2 KP.3.11 | cell entry | Dadonaite et al. (2025b) |
| SCV2 KP.3.11 | binding | Dadonaite et al. (2025b) |
| IAV H3N2 HK19 | immune escape | Welsh et al. (2024) |
| IAV H3N2 HK19 | viral growth | Welsh et al. (2024) |
| IAV H5N1 | immune escape | Dadonaite et al. (2024a) |
| IAV H5N1 | cell entry | Dadonaite et al. (2024a) |
| IAV H5N1 | stability | Dadonaite et al. (2024a) |
| IAV H3N2 MC22 | immune escape | Yu et al. (2025) |
| IAV H3N2 MC22 | cell entry | Yu et al. (2025) |
| IAV H3N2 MC22 | stability | Yu et al. (2025) |
| *IAV H1N1 | viral growth | Doud & Bloom (2016) |
| *IAV H1N1 | viral growth | Wu et al. (2014) |
| *IAV H2N1 | viral growth | Soh et al. (2019) |
| *IAV H3N2 | viral growth | Lee et al. (2018) |
| *IAV H3N2 | viral growth | Doud et al. (2015) |
| *IAV H1N1 | viral growth | Doud et al. (2015) |
| *IAV H1N1 | viral growth | Jiang et al. (2016) |
| *IAV H1N1 | viral growth | Wu et al. (2015) |
| *CXB3 | viral growth | Mattenberger et al. (2021) |
| *AAV2 | viral growth | Sinai et al. (2021) |
| *DEN | viral growth | Suphatrakul et al. (2023) |
| *HCV JFH 1 | viral growth | Qi et al. (2014) |
| *PESV | stability | Tsuboyama et al. (2023) |

*represents the dataset is from ProteinGym

Table 7: Distribution of viruses and phenotypes (total counts) for DMS functional assays.

| | Binding | Cell entry | Expression | Fitness | Immune escape | Stability | Viral growth | Total |
|---|---|---|---|---|---|---|---|---|
| AAV2 | | | | | | | 1 | 1 |
| CXB3 | | | | | | | 1 | 1 |
| DEN | | | | | | 1 | | 1 |
| HBV | | | | 1 | | | | 1 |
| HCV | | | | | | | 1 | 1 |
| HIV | | 2 | | | 3 | | 6 | 11 |
| IAV | | 2 | | | 3 | 2 | 9 | 16 |
| LASV | | 1 | | | 1 | | | 2 |
| NIPAH | 1 | 1 | | | 1 | | | 3 |
| PESV | | | | | | 1 | | 1 |
| RABV | | 1 | | | 1 | | | 2 |
| SCV2 | 15 | 4 | 12 | | 4 | | 2 | 37 |
| ZIKV | | | | | 1 | | 1 | 2 |
| Total | 16 | 11 | 12 | 1 | 14 | 3 | 22 | 79 |

Table 8: Sources of neutralisation benchmark datasets. Antigenicity diversity prediction tasks are based on neutralisation assays. This table summarise the data resources from publicly available datasets.

| Dataset | Sera Source | Subtype | Reference |
|---|---|---|---|
| A/Massachusetts/18/2022 | ferret | H3N2 | Kikawa et al. (2025a) |
| A/Thailand/8/2022 | ferret | H3N2 | Kikawa et al. (2025a) |
| A/DistrictOfColumbia/27/2023 | ferret | H3N2 | Kikawa et al. (2025a) |
| A/Croatia/10136RV/2023-egg | ferret | H3N2 | Kikawa et al. (2025a) |
| A/Netherlands/10563/2023 | ferret | H3N2 | Kikawa et al. (2025a) |
| A/Lisboa/216/2023 | ferret | H3N2 | Kikawa et al. (2025a) |
| A/Slovenia/49/2024 | ferret | H3N2 | Kikawa et al. (2025a) |
| A/Switzerland/47775/2024 | ferret | H3N2 | Kikawa et al. (2025a) |
| A/Norway/12374/2023 | ferret | H3N2 | Kikawa et al. (2025a) |
| A/BurkinaFaso/3131/2023 | ferret | H3N2 | Kikawa et al. (2025a) |
| A/France/IDF-IPP29542/2023-egg | ferret | H3N2 | Kikawa et al. (2025a) |
| A/Netherlands/10685/2024 | ferret | H3N2 | Kikawa et al. (2025a) |
| A/Lisboa/188/2023 | ferret | H1N1 | Kikawa et al. (2025a) |
| A/Victoria/4897/2022 | ferret | H1N1 | Kikawa et al. (2025a) |
| A/Victoria/4897/2022_IVR-238 | ferret | H1N1 | Kikawa et al. (2025a) |
| A/Wisconsin/67/2022 | ferret | H1N1 | Kikawa et al. (2025a) |
| A/Norway/07606/2024 | ferret | H1N1 | Kikawa et al. (2025a) |
| A/Tajikistan/02-1057/2024 | ferret | H1N1 | Kikawa et al. (2025a) |
| A/Darwin/9/2021 | human | H3N2 | Kikawa et al. (2025b) |
| A/Massachusetts/18/2022 | human | H3N2 | Kikawa et al. (2025b) |
| A/Wisconsin/588/2019 | human | H1N1 | Loes et al. (2024) |

## A.2 EXTENDED RESULTS

Table 9: Results for all models across tasks.

| Model Name | DMS | | | | Neutralisation | | GISAID | |
| | Recall | Std. | Spearman | Std. | Spearman | Std. | Recall | Spearman |
| --- | --- | --- | --- | --- | --- | --- | --- | --- |
| VESPAl | 0.1635 | 0.0726 | 0.2715 | 0.1402 | 0.1961 | 0.206 | 0.3226 | **0.3602** |
| VESPA | 0.1702 | 0.0796 | 0.2797 | 0.1506 | 0.1961 | 0.206 | 0.2936 | 0.3152 |
| Tranception S | 0.1358 | 0.0395 | 0.1776 | 0.1124 | 0.1843 | 0.1810 | 0.1080 | 0.0874 |
| Tranception M | 0.1530 | 0.0595 | 0.2271 | 0.1300 | **0.2316** | 0.1696 | 0.1579 | 0.1456 |
| Tranception L | 0.1572 | 0.0681 | 0.2164 | 0.1296 | 0.1927 | 0.1961 | 0.2066 | 0.2196 |
| ProtGPT2 | 0.1105 | 0.0370 | 0.1021 | 0.0732 | 0.2018 | 0.1845 | 0.1221 | 0.0372 |
| ProGen2 | 0.1873 | 0.0823 | 0.2825 | 0.1644 | 0.2018 | 0.1929 | 0.3790 | 0.3235 |
| ProGen2 S | 0.1592 | 0.0671 | 0.2433 | 0.1596 | 0.2250 | 0.1852 | 0.1944 | 0.1955 |
| ProGen2 M | 0.1838 | 0.0829 | 0.2884 | 0.1701 | 0.2240 | 0.2070 | 0.2636 | 0.2414 |
| ProGen2 OAS | 0.0972 | 0.0231 | 0.0392 | 0.0367 | 0.1829 | 0.1513 | 0.0970 | 0.0746 |
| ProGen2 BFD90 | 0.1888 | 0.0802 | 0.2798 | 0.1563 | 0.2147 | 0.1981 | 0.3798 | 0.2933 |
| ProGen2 L | 0.1771 | 0.0727 | 0.2626 | 0.1553 | 0.2101 | 0.2002 | 0.2577 | 0.2519 |
| ProGen2 XL | **0.1980** | 0.0910 | **0.2930** | 0.1583 | 0.2093 | 0.1977 | **0.4081** | 0.3153 |
| ESM1v | 0.1451 | 0.0644 | 0.1877 | 0.1026 | 0.2282 | 0.2043 | 0.1206 | 0.0543 |
| ESM1 43M | 0.1454 | 0.0655 | 0.1901 | 0.0988 | 0.2222 | 0.2098 | 0.1858 | 0.1384 |
| ESM1 85M | 0.1416 | 0.0531 | 0.1799 | 0.0918 | 0.2024 | 0.2217 | 0.2471 | 0.1870 |
| ESM1 670M UR50S | 0.1466 | 0.0586 | 0.1997 | 0.1021 | 0.1957 | 0.1943 | 0.1956 | 0.1533 |
| ESM1 670M UR50D | 0.1387 | 0.0575 | 0.1792 | 0.0940 | 0.2093 | 0.2039 | 0.1921 | 0.1420 |
| ESM1 670M UR100 | 0.1272 | 0.0512 | 0.1080 | 0.0963 | 0.2029 | 0.1725 | 0.2251 | 0.2307 |
| ESM2 8M | 0.1280 | 0.0514 | 0.1401 | 0.0852 | 0.2053 | 0.1939 | 0.2368 | 0.2893 |
| ESM2 35M | 0.1289 | 0.0512 | 0.1417 | 0.0956 | 0.1961 | 0.1985 | 0.2608 | 0.3026 |
| ESM2 150M | 0.1307 | 0.0531 | 0.1156 | 0.0770 | 0.2174 | 0.2014 | 0.2141 | 0.2023 |
| ESM2 650M | 0.1375 | 0.0616 | 0.1693 | 0.1040 | 0.2267 | 0.1840 | 0.2333 | 0.2239 |
| ESM2 3B | 0.1419 | 0.0639 | 0.1672 | 0.1066 | 0.2148 | 0.1889 | 0.1889 | 0.2075 |
| ESM2 15B | 0.1390 | 0.0716 | 0.1741 | 0.1122 | 0.2039 | 0.1923 | 0.2451 | 0.1851 |

Table 10: Results across 57 non-indel mutational effect tasks. Improvement is calculated as the relative percentage change over previous results.

| Model Name | Recall | Improvement (%) | Spearman | Improvement (%) |
|---|---|---|---|---|
| ESM2 3B | 0.1483 | 4.5102 | 0.1885 | 12.7392 |
| ESM1 670M UR50D | 0.1446 | 4.2538 | 0.1851 | 3.2924 |
| Tranception L | 0.1632 | 3.8168 | 0.2263 | 4.5749 |
| VESPA | 0.1764 | 3.6428 | 0.2983 | 6.6500 |
| ProGen2 OAS | 0.1006 | 3.4979 | 0.0453 | 15.5612 |
| ESM2 650M | 0.1412 | 2.6909 | 0.1832 | 8.2103 |
| ESM1v | 0.1487 | 2.4810 | 0.2023 | 7.7784 |
| VESPA1 | 0.1675 | 2.4465 | 0.2832 | 4.3094 |
| ProGen2 XL | 0.2025 | 2.2727 | 0.2979 | 1.6724 |
| ESM2 8M | 0.1304 | 1.8750 | 0.1518 | 8.3512 |
| ESM2 15B | 0.1416 | 1.8705 | 0.1788 | 2.6996 |
| Tranception M | 0.1557 | 1.7647 | 0.2234 | -1.6292 |
| ProGen2 L | 0.1800 | 1.6375 | 0.2707 | 3.0845 |
| ProGen2 M | 0.1868 | 1.6322 | 0.2891 | 0.2427 |
| ESM1 43M | 0.1477 | 1.5818 | 0.1893 | -0.4208 |
| ESM1 85M | 0.1432 | 1.1299 | 0.1815 | 0.8894 |
| ProGen2 BFD90 | 0.1901 | 0.6886 | 0.2796 | -0.0715 |
| ESM1 670M UR50S | 0.1471 | 0.3411 | 0.2028 | 1.5523 |
| Tranception S | 0.1359 | 0.0736 | 0.1753 | -1.2950 |
| ESM2 150M | 0.1304 | -0.2295 | 0.1537 | 32.9585 |
| ESM1 670M UR100 | 0.1259 | -1.0220 | 0.0986 | -8.7037 |
| ProtGPT2 | 0.1086 | -1.7195 | 0.0949 | -7.0519 |
| ProGen2 S | 0.1560 | -2.0101 | 0.2402 | -1.2741 |
| ProGen2 | 0.1835 | -2.0288 | 0.2764 | -2.1593 |
| ESM2 35M | 0.1216 | -5.6633 | 0.1325 | -6.4926 |

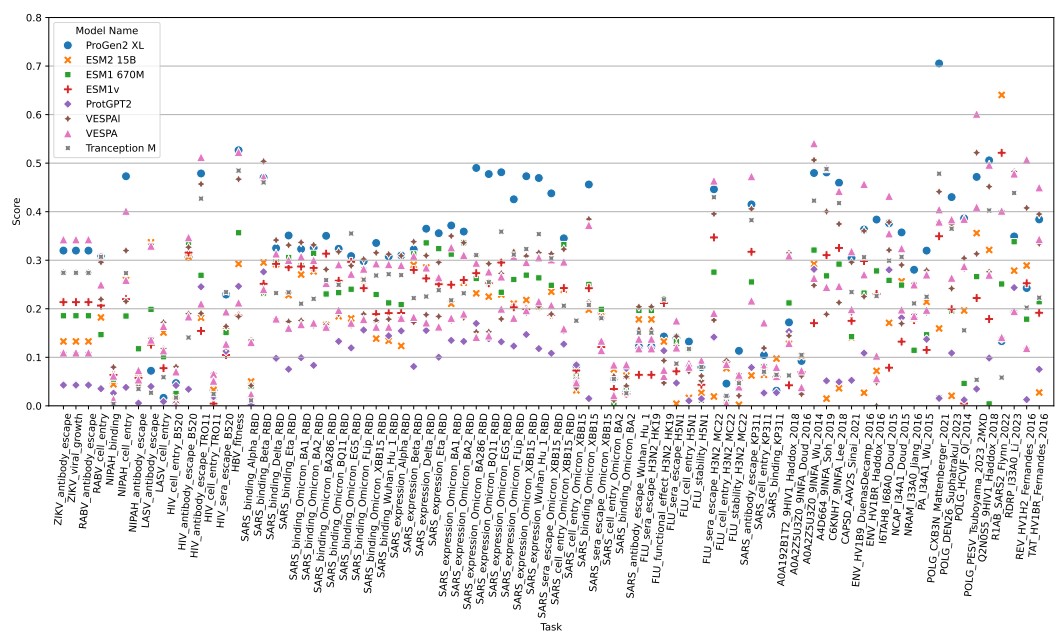

Figure 6: Task-wise comparison of all baselines on the DMS benchmark. Reported values represent the absolute Spearman's rank correlation between model fitness scores and experimental measurements.

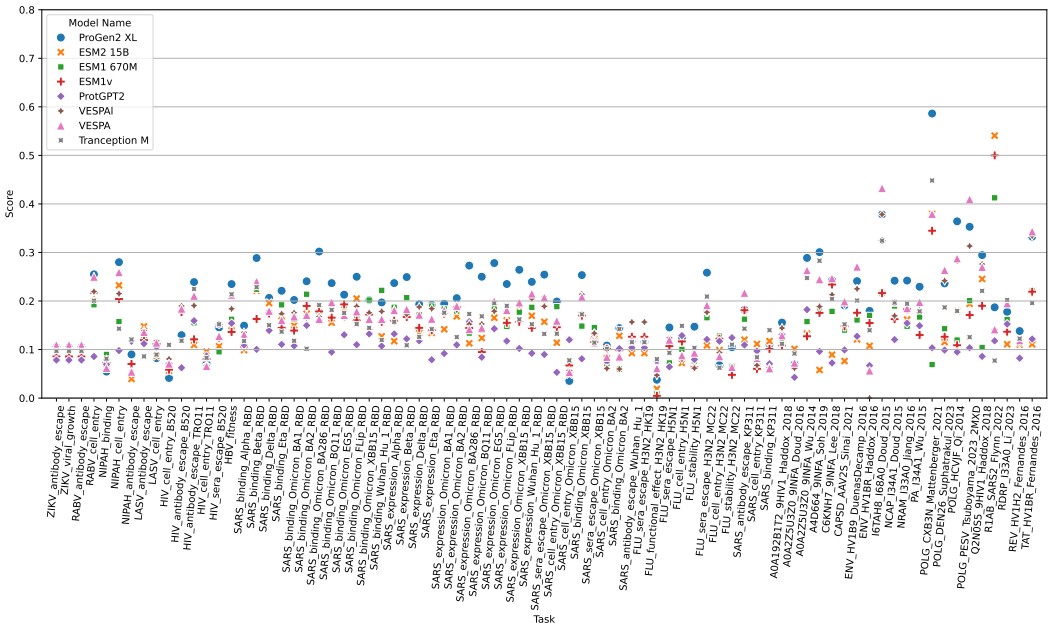

Figure 7: Task-wise comparison of all baselines on the DMS benchmark. Reported values represent the top 10% recall between model fitness scores and experimental measurements.

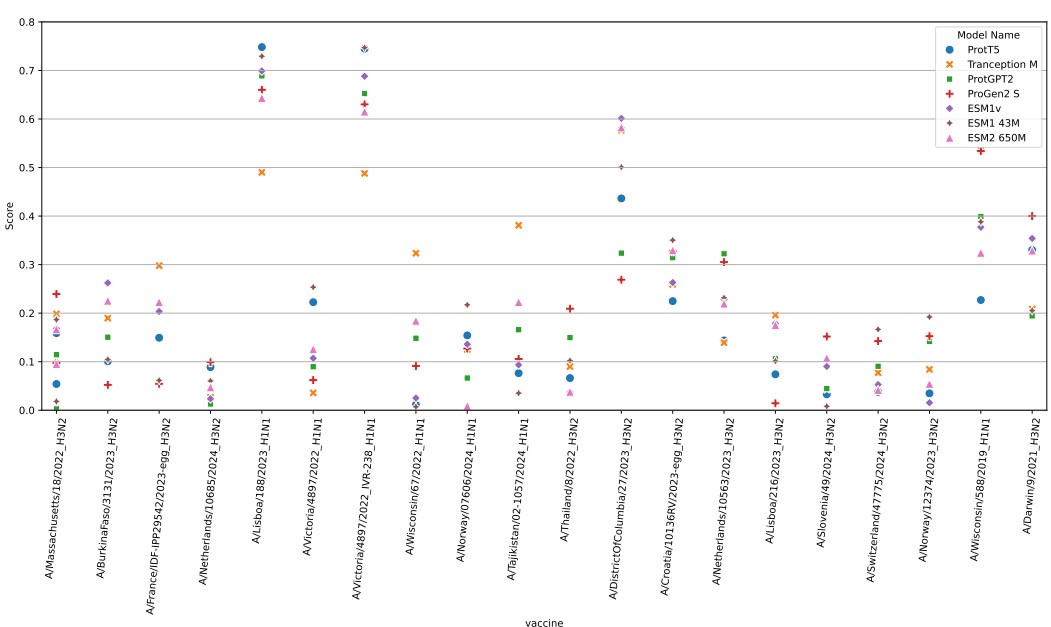

Figure 8: Task-wise comparison of all baselines on the neutralisation benchmark. Reported values represent the absolute Spearman's rank correlation between model fitness scores and experimental measurements.

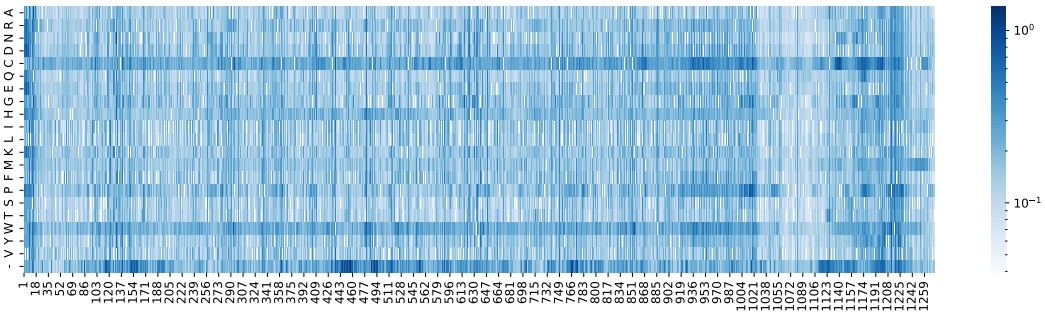

Figure 9: SARS-CoV-2 Spike Protein Mutation Heat Map for ESM1. This heat map displays the frequency of 21 potential amino acid substitutions across 1273 residues of the SARS-CoV-2 Spike protein, with colour intensity indicating mutational effect at each position.

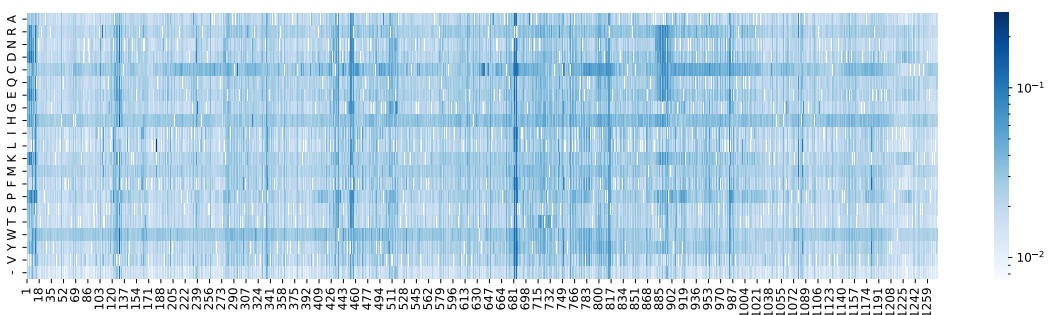

Figure 10: SARS-CoV-2 Spike Protein Mutation Heat Map for ESM2. This heat map displays the frequency of 21 potential amino acid substitutions across 1273 residues of the SARS-CoV-2 Spike protein, with colour intensity indicating mutational effect at each position.

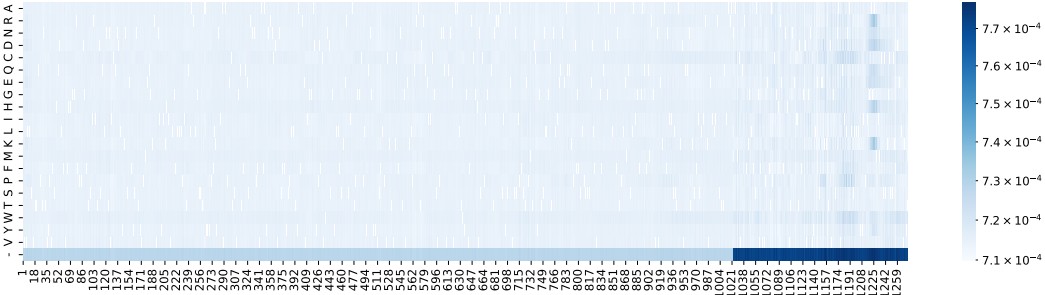

Figure 11: SARS-CoV-2 Spike Protein Mutation Heat Map for ESM1v. This heat map displays the frequency of 21 potential amino acid substitutions across 1273 residues of the SARS-CoV-2 Spike protein, with colour intensity indicating mutational effect at each position.

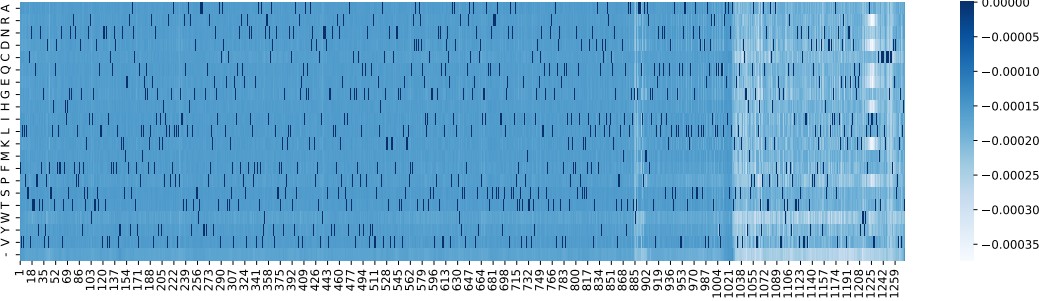

Figure 12: SARS-CoV-2 Spike Protein Mutation Heat Map for ProGen2. This heat map displays the frequency of 21 potential amino acid substitutions across 1273 residues of the SARS-CoV-2 Spike protein, with colour intensity indicating mutational effect at each position.

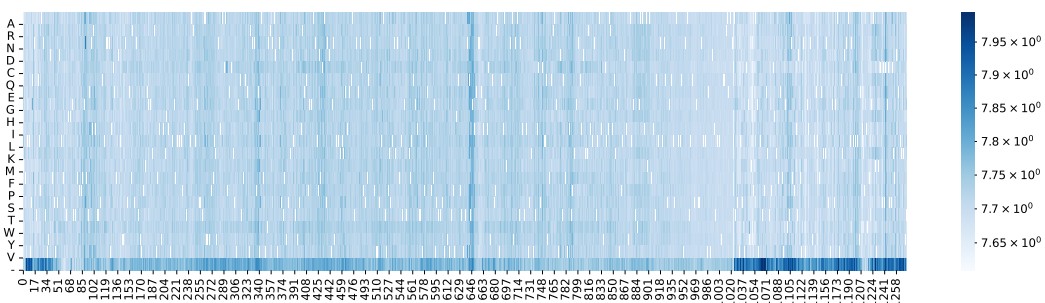

Figure 13: SARS-CoV-2 Spike Protein Mutation Heat Map for ProtGPT2. This heat map displays the frequency of 21 potential amino acid substitutions across 1273 residues of the SARS-CoV-2 Spike protein, with colour intensity indicating mutational effect at each position.

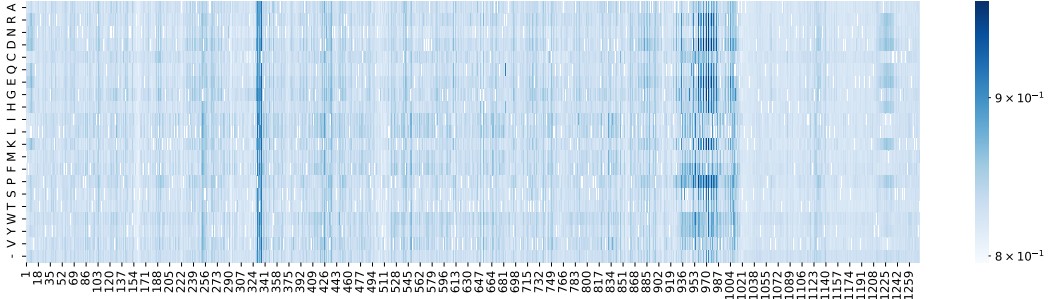

Figure 14: SARS-CoV-2 Spike Protein Mutation Heat Map for Tranception. This heat map displays the frequency of 21 potential amino acid substitutions across 1273 residues of the SARS-CoV-2 Spike protein, with colour intensity indicating mutational effect at each position.

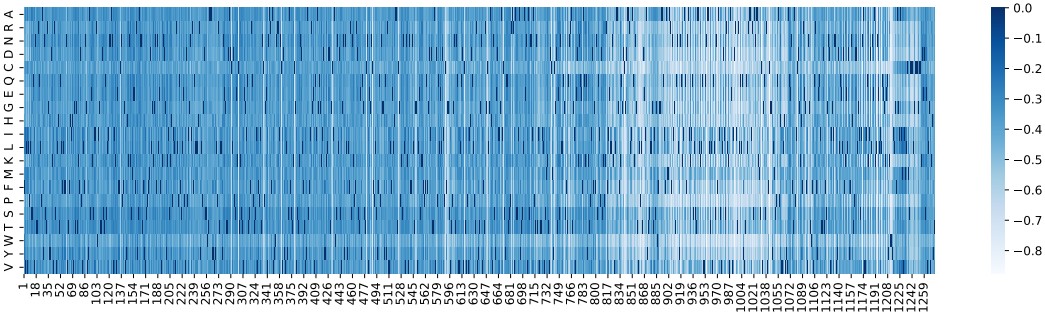

Figure 15: SARS-CoV-2 Spike Protein Mutation Heat Map for VESPA. This heat map displays the frequency of 21 potential amino acid substitutions across 1273 residues of the SARS-CoV-2 Spike protein, with colour intensity indicating mutational effect at each position.

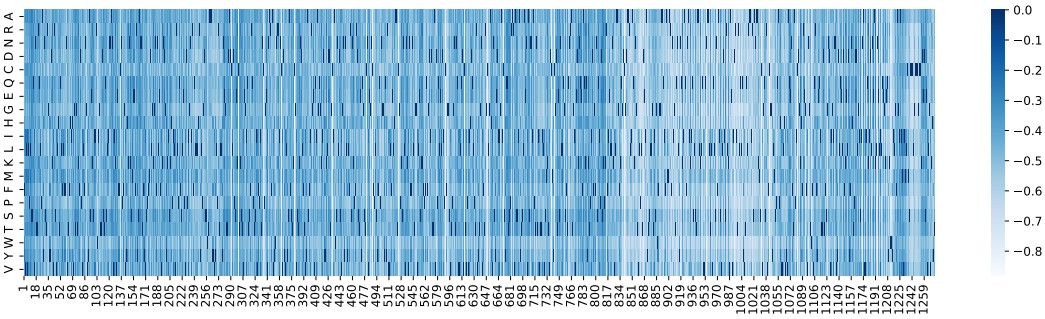

Figure 16: SARS-CoV-2 Spike Protein Mutation Heat Map for VESPAl. This heat map displays the frequency of 21 potential amino acid substitutions across 1273 residues of the SARS-CoV-2 Spike protein, with colour intensity indicating mutational effect at each position.

