# OpenReview forum: "ViroGym: Realistic Large-Scale Benchmarks for Evaluating Viral Proteins"
_ICLR.cc/2026/Workshop/LMRL — Submitted to ICLR 2026 Workshop LMRL_

### Official Review · Reviewer_aTUp · 2026-02-20
**A viral protein benchmark with broad data curation but weak analytical depth and limited methodological novelty.**

**Rating:** 5
**Confidence:** 4

**Review:**

ViroGym is a benchmark for evaluating protein language models (pLMs) on viral protein tasks. It curates 79 DMS assays (552,937 mutated sequences across 13 virus types and 7 phenotypic categories), 21 influenza neutralization assays, and a SARS-CoV-2 pandemic forecasting task using GISAID mutation frequencies. The paper benchmarks ESM-1/2, ProtT5, ProGen2, Tranception, ProtGPT2, and VESPA/VESPAl in zero-shot settings, finding that ProGen2-XL performs best overall and that its top predicted mutations overlap ~50% with dominant circulating SARS-CoV-2 mutations.

Scale and breadth of curation. 79 DMS assays across 13 virus types with 7 phenotypic categories represents a meaningful expansion over ProteinGym's 24 viral assays and EVEREST's 45 datasets. The inclusion of immune escape as a distinct phenotype category is well-motivated for vaccine applications. The detailed provenance tables (Tables 5–8) enable reproducibility.
Three-tier evaluation design. The progression from controlled DMS experiments → neutralization assays → real-world GISAID surveillance is conceptually appealing and reflects a genuine translational pipeline. The finding that DMS-top mutations have only 10% overlap with circulating variants while ProGen2-XL achieves ~50% overlap (Figure 5) is an interesting and potentially impactful observation.
Practical relevance. The connection to vaccine strain selection is well-motivated, timely, and represents a genuine unmet need. The neutralization assay component (21 tasks with curated vaccination histories) is a unique contribution not present in other benchmarks.
Comprehensive model coverage. 25+ model variants are evaluated (Table 9), including different sizes of ESM-2, ProGen2, and Tranception families, enabling scaling analysis.
Semantic scoring finding. The observation that Euclidean distance between mean-pooled embeddings outperforms masked/wildtype/mutation marginals for encoder models (Table 1) is a useful practical finding, though it requires more rigorous validation.

Benchmark construction lacks novelty beyond aggregation. The core contribution is assembling existing public DMS datasets and running existing pLMs with existing scoring strategies. The DMS curation follows ProteinGym guidelines. The models are all published. The scoring strategies are all published. The metrics (Spearman, Recall@K) are standard. This is useful infrastructure work, but the paper does not introduce any new methods, metrics, or analytical frameworks beyond compilation.
Performance levels are alarmingly poor across the board, and this is underanalyzed. The best DMS Spearman correlation is 0.293 (ProGen2-XL) — meaning the best model explains less than 9% of variance in mutational effects. Top 10% recall of 0.198 means 80% of the most impactful mutations are missed. The neutralization Spearman values hover around 0.20–0.23 with standard deviations nearly as large as the means. Rather than deeply analyzing why performance is so poor (which mutations/viruses/phenotypes are hardest? what structural or evolutionary features predict failure?), the paper largely presents tables and moves on. A benchmark paper's value is proportional to the insights it generates about model failures, and these are largely absent.
The pandemic prediction task has critical methodological flaws. (a) The GISAID mutation frequency is compared against pLM fitness scores, but mutation frequency in circulating viruses reflects transmission fitness, immune escape, founder effects, and surveillance bias — not just protein function. Equating frequency with "fitness" conflates distinct evolutionary forces. (b) The top-10 mutation overlap analysis (Figure 5) uses an extremely small sample (10 mutations) where chance overlap is non-negligible. No null model or statistical test is provided. With 1273 × 20 = 25,460 possible mutations, picking 10 at random would yield ~0.4% overlap; the observed overlaps should be tested against this baseline. (c) Precision@3 (Table 4) has such high variance at K=3 that only ProGen2-XL achieves a non-zero value (0.33, i.e., 1 correct out of 3), which is hardly robust evidence.
Neutralization task evaluation is superficial. The paper computes embedding distance between vaccine strain and circulating strain as "predicted antigenic distance," but this collapses the complex antibody–antigen interaction landscape into a single number. No ablation is performed (e.g., distance computed over just the HA1 head domain vs. full sequence? Layer-wise analysis?). The Spearman correlations in Table 3 are weak (~0.20) and statistically indistinguishable across models — the paper acknowledges this but draws no deeper insight.
The semantic scoring claim (Section 4.1) is inadequately supported. Table 1 compares 7 scoring strategies for ESM2-650M only. The claim that "semantic" (Euclidean embedding distance) is "the most effective strategy" is based on a Spearman of 0.169 vs. 0.109 for masked marginals. But: (a) this is tested on one model, (b) the standard deviations overlap substantially (0.104 vs. 0.121), and (c) no statistical test is performed. Yet the paper uses semantic scoring as the default for all encoder models throughout, potentially disadvantaging or advantaging specific models depending on how this choice interacts with architecture.
Missing critical baselines and analytical dimensions. (a) No MSA-based methods (EVE, EVEscape) are benchmarked. The paper justifies this by noting MSAs are "difficult to obtain for novel viruses," which is true for truly novel viruses but not for SARS-CoV-2 or influenza, which have massive sequence databases. Since these are the primary viruses in ViroGym, this exclusion is not well-justified. (b) No structure-based methods (ESMFold, AlphaFold-derived features) despite their increasing availability. (c) No analysis of how phylogenetic distance between the target virus and training data distribution affects performance — this is arguably the single most important confound for pLM evaluation on viral proteins. (d) No per-virus or per-phenotype breakdown in the main text; the reader must dig through appendix figures to find task-level results.

---

### Official Review · Reviewer_Ye8i · 2026-02-24
**Concerns Regarding the Methodology**

**Rating:** 3
**Confidence:** 5

**Review:**

This paper introduces ViroGym, a large-scale benchmark for evaluating protein language models (pLMs) specifically on viral proteins. The benchmark integrates:
	1.	79 DMS assays (552,937 mutated sequences across 13 viruses and 7 phenotypes),
	2.	21 influenza neutralization tasks, and
	3.	A real-world pandemic forecasting task based on SARS-CoV-2 mutation frequencies from GISAID

While this addresses an important problem, the GISAID “pandemic prediction” task is technically questionable. The authors use single-mutation frequency in sequenced genomes as a proxy for fitness, implicitly assuming that fitness is approximated by additive single-mutation effects. However, real SARS-CoV-2 evolution is driven by multi-mutation haplotypes, strong epistatic interactions, and background-dependent effects, meaning that single-mutation frequency does not accurately capture lineage-level transmission advantage or evolutionary fitness. A more scientifically grounded evaluation would model epidemiological fitness using growth curve (e.g., approaches similar to Science 2022, doi:10.1126/science.abm1208), rather than relying on marginal mutation counts alone. Based on these concerns, I am worried that the proposed benchmark may inadvertently mislead the AI field for viral forecasting and pandemic prevention.

In addition, while the viral dms benchmark is broader in scope (80 viral DMS), it is not clear how it meaningfully advances beyond prior viral-specific benchmarks such as EVEREST (45 viral DMS) datasets since no new methodology is introduced.  The primary novelty appears to be the mutation-frequency forecasting component, but this is precisely the most methodologically problematic element.

The manuscript also does not sufficiently engage with recent literature on viral escape forecasting beyond the Marks lab. To develop a benchmark intended to guide viral forecasting, the authors should situate their benchmark relative to recent deep learning and evolutionary modeling approaches (e.g., Nature Communications 2025, doi:10.1038/s41467-025-59422-w; PNAS 10.1073/pnas.2503742122; PLoS Computational Biology 10.1371/journal.pcbi.1013582).

Last but not least, the manuscript does not address biosecurity considerations. By curating large-scale datasets across multiple viral pathogens—including SARS-CoV-2, Influenza A (H1N1, H3N2, H5N1), HIV, Zika virus, Rabies virus, Nipah virus, Lassa virus, Hepatitis B and C viruses, Dengue virus etc, the benchmark introduces clear dual-use implications. A dedicated biosecurity and responsible-use assessment is warranted. see relevant (https://www.nature.com/articles/s41587-025-02650-8).

---

### Meta-Review · Area_Chair_XjPT · 2026-02-25

**Recommendation:** Reject
**Confidence:** 4

**Metareview:**

I recommend rejection.

---

### Decision · Program_Chairs · 2026-03-02

**Decision:**

Reject

**Comment:**

Please see the meta-review.